# Developmental Hurdles That Can Compromise Pregnancy during the First Month of Gestation in Cattle

**DOI:** 10.3390/ani13111760

**Published:** 2023-05-25

**Authors:** Savannah L. Speckhart, Mary A. Oliver, Alan D. Ealy

**Affiliations:** School of Animal Science, Virginia Tech, Blacksburg, VA 24061, USA; slspeckhart@vt.edu (S.L.S.); maryalio@vt.edu (M.A.O.)

**Keywords:** pregnancy, embryo, in vitro embryo production, bovine

## Abstract

**Simple Summary:**

A high incidence of early pregnancy failures exists in beef and dairy cows. This review describes various problems which occur during early gestation that contribute to these pregnancy losses. This review also describes why in-vitro-produced bovine embryos exhibit a greater susceptibility to pregnancy failure. These pregnancy losses are economically costly. More research is needed to identify ways to limit early pregnancy losses.

**Abstract:**

Several key developmental events are associated with early embryonic pregnancy losses in beef and dairy cows. These developmental problems are observed at a greater frequency in pregnancies generated from in-vitro-produced bovine embryos. This review describes critical problems that arise during oocyte maturation, fertilization, early embryonic development, compaction and blastulation, embryonic cell lineage specification, elongation, gastrulation, and placentation. Additionally, discussed are potential remediation strategies, but unfortunately, corrective actions are not available for several of the problems being discussed. Further research is needed to produce bovine embryos that have a greater likelihood of surviving to term.

## 1. Introduction

A successful cow pregnancy relies on the achievement of several events within the first month of gestation. Unfortunately, approximately 45% of suckling beef and lactating cows will be nonpregnant within the first month following a single insemination [1,2]. Errors in one of several key developmental events can lead to pregnancy loss. These key events include fertilization (day 0), genome activation (day 3–4), blastocyst development (day 7–8), blastocyst hatching (day 8–10), initiation of gastrulation (day 11–12) and elongation (day 13–14), maternal recognition of pregnancy (day 15–17), initial attachment to the uterine wall (day 19–20), and chorioallantoic placenta formation (day 30–40) [3,4]. These and other mishaps occurring in the first month of gestation may occur because of problems with the embryo itself, or it may result from insufficiencies in oviductal and uterine function. We will not attempt to distinguish between inherent embryo-derived losses and maternal-derived losses as this has been addressed nicely by other authors recently [3,4,5]. Instead, we will focus on explaining the etiology of the various developmental issues that may cause pregnancy loss. Pregnancy losses continue after this time, and these losses have substantial financial consequences to dairy and beef producers [2,6], but this review will focus our discussions on exploring how mishaps in embryogenesis events occurring during the first month of gestation may influence pregnancy loss in cattle. 

Another emphasis of this review is to examine how the progression of these key developmental events is influenced when bovine embryos are generated through in vitro maturation, fertilization, and culture. These in-vitro-produced (IVP) embryos resemble and develop like in-vivo-derived embryos in some regards, but notable differences in transcript profiles, epigenetic patterns, and cell numbers and types exist for IVP embryos (see [3,7,8] for recent reviews). These discussion points are especially relevant for two reasons. First, IVP bovine embryos are an exceptionally good model for exploring the root causes of pregnancy loss during early embryogenesis in cattle. Second, there is an ever-growing commercial usage for IVP embryos. The dairy and beef industries are combining embryo systems using ovum pickup (OPU) techniques to generate large numbers of embryos from genetically elite heifers/cows and bulls [9]. In 2021, embryos derived from OPU-IVP accounted for over one-million embryos used in embryo transfer (ET) globally [10]. This was three-times greater than the number of in-vivo-generated embryos used for ET [10]. 

Unfortunately, IVP embryos remain inferior to in-vivo-generated embryos in several regards. Only 25–40% of fertilized oocytes will reach the blastocyst stage, the ideal stage for transfer [8,9]. Some early embryonic death is warranted given that some embryos contain chromosomal abnormalities, but it may be possible that a portion of this poor in vitro development potential can one day be overcome with continued refinements in IVP embryo systems. Another shortcoming for IVP embryos is that their pregnancy retention rates are routinely 10–20% less than those observed after artificial insemination or ET using in-vivo-derived embryos [3,9]. Most of the pregnancy losses experienced by IVP embryos occurs during the first month of gestation [3]. There has been a long-standing concern that IVP embryos generate calves with developmental deformities and reduced lifetime performance [11,12,13]. Issues with calf birth weight seem to have been solved by removing serum from media formulations and culturing embryos in refined media formulations under hypoxic conditions [11]. Recent reports in the past ten years suggest that neonatal calf survival may still be compromised in IVP pregnancies [13,14], but most IVP-generated calves produce an equivalent amount of milk as their herd mates, although they do experience a slight decrease in fertility [15]. 

Each of the primary key factors that limit the developmental potential of in-vivo- and in-vitro-generated embryos and pregnancies will be examined in the following sections. Summaries of key limitations and remediation strategies are provided in Table 1 and Table 2. Many of the discussion points will focus primarily on the bovine embryo and conceptus, but we will also discuss facets of embryology in other species when they are germane. 

## 2. Oocyte Maturation

A mature and competent oocyte is a requisite to the success of early embryogenesis. There are two categories of oocyte maturation: nuclear and cytoplasmic. The resumption of meiosis is a key feature of oocyte nuclear maturation [16]. The first step in achieving that milestone is germinal vesicle breakdown (GVBD), which is initiated by a release of luteinizing hormone (LH) from the pituitary gland [17,18]. The timing of GVBD is species-specific and requires 6 to 8 h to occur in cattle [17]. However, GVBD alone is not sufficient to induce meiotic resumption. High intrafollicular levels of cyclic adenosine monophosphate (cAMP) maintains high protein kinase A (PKA) activity, and this phosphorylates critical cell cycle components which prevents the activation of M-phase promoting factor (MPF) [18,19,20]. Elevated cAMP levels is facilitated by help from neighboring granulosa cells producing cyclin guanosine monophosphate (cGMP), an inhibitor of phosphodiesterase 3A (PDE3A), the main oocyte phosphodiesterase responsible for hydrolyzing cAMP [21]. The transport of cGMP from granulosa cells to the oocyte is made possible by gap junctions [22]. 

Nuclear maturation is initiated with the surge in luteinizing hormone (LH). One of the primary actions of LH is to interrupt cumulus cell and oocyte communication for meiosis resumption. This cell-to-cell communication is mediated by Connexin 43 (Cx43) channels that create gap junctions between granulosa and cumulus cells as well as between cumulus cells and the oocyte [23]. The surge in LH also reduces cumulus cell cGMP levels, and this activates PDE3A, thereby reducing cAMP blockage of MPF [24]. Interestingly, the LH receptor is a G-protein-coupled receptor that activates adenylyl cyclase, leading to increases in the intracellular cAMP concentrations within granulosa cells [21,25]. This is intriguing because an increase in cAMP within granulosa and cumulus cells is needed to drive down cAMP in the oocyte. Activation of the epidermal growth factor receptor (EGFR) is another crucial component in triggering LH-induced meiotic resumption. The rapid release of EGFR ligands, epiregulin and amphiregulin, from granulosa and cumulus cells is an important catalyst of LH receptor activation [21,25]. The EGFR kinase-dependent components rapidly drive down the cGMP concentrations in these cells. This maturation process also will ensue if there is loss of competent gap junctions or when the oocyte is forcibly removed from its follicular environment either because of spontaneous ovulation or when being placed into in vitro maturation (IVM) conditions. Various supplemental, membrane-permeable cAMP agonists can effectively delay nuclear maturation, and maturation ensues quickly after removing these compounds [26,27,28,29,30]. These procedures are beneficial for various experimental purposes, including work focused on the cytoplasmic maturation of oocytes. 

Cytoplasmic maturation involves the reorganization of organelles and cortical granules within the cytoplasm [31,32,33]. Following GVBD, the assembly of the DNA spindle is adjacent to the newly released first polar body along the edge of the ooplasm. Interestingly, the germinal vesicle position in mouse oocytes is a predictor of oocyte developmental competence [34], and the age of the donor influences the germinal vesicle position [35]. The reorganization of mitochondria to areas of high energy consumption is essential for the oocytes and embryo blastomere during critical points in the cell cycle [36,37]. The oocyte mitochondria move from a peripheral position to a more dispersed distribution throughout the cytoplasm around 12–18 h into maturation [33]. After reaching metaphase II (MII), the mitochondria along with lipid droplets maintain a central position in the ooplasm where they apparently are primed to metabolize stored triglycerides, providing the zygote with energy until embryonic genome activation occurs [33,36,37]. Another organelle that is reorganized are cortical granules, which are derived from the Golgi complex. Oocytes in the GV stage have distributed clusters of cortical granules throughout the cytoplasm (Wang et al., 1997). Once oocytes reach the MII stage, the granules are distributed close to the plasma membrane where they wait to be released during fertilization to prevent polyspermy [32,33].

Another very important aspect of cytoplasmic maturation is the influx of proteins, ribonucleotides, lipids, and other agents from cumulus cells [18,38]. These molecules are essential for maturation and for the progression of the early embryo up to the 8- to 16-cell stage in cattle, when the embryonic genome is activated [7]. mRNAs and proteins have been the topic of much investigation given that they are needed to coordinate the complex array of processes and functions that take place in the zygote and embryo for the first few days of development. An excellent recent review describes the coordinated accumulation, storage, usage, and degradation of these mRNAs and proteins, so this topic will not be discussed further here [7]. 

Cumulus expansion is commonly used as a metric of oocyte maturation, but this does not provide a highly accurate assessment of the efficiency of nuclear and cytoplasmic maturation. One strategy that is being used to is to use brilliant cresyl blue (BCB) staining as a marker of oocyte competence [39,40,41,42,43]. This cell-permeable, non-toxic stain is an indicator of cytoplasmic maturation. This is made possible by indicating the level of glucose-6-phosphate dehydrogenase (G6PDH) activity, which is low in oocytes that have undergone cytoplasmic maturation [44]. Another way to improve IVM success is by adding growth factors, cytokines, and other bioactive factors that are normally present within the follicle niche but may be absent in our in vitro maturation systems. Most culture system supplement FSH, which usually also contains LH, and estradiol during IVM to promote cumulus expansion and meiotic progression to MII [45,46]. Some bovine IVM media formulations also include epidermal growth factor (EGF). This supplement enhances the biological effects of FSH and LH by mimicking the actions of EGF-like molecules, amphiregulin and epiregulin, that are normally produced by cumulus cells and act via EGFR [47,48,49]. Indeed, increases in oocyte maturation and blastocyst development rates have been observed by including EGF in IVM medium [50,51,52,53,54,55]. A mixture of fibroblast growth factor 2 (FGF2), leukemia inhibitory factor (LIF), and insulin-like growth factor 1 (IGF1) has also been explored for its ability to improve oocyte competency and subsequent blastocyst development in cattle. Supplementing this mixture, abbreviated as “FLI”, during porcine oocyte maturation greatly improved oocyte maturation and subsequent blastocyst yields [56], but these benefits were not observed in bovine oocytes [57]. More work is needed to identify novel factors and combinations of factors that can improve bovine IVP efficiency. 

Another interesting way to potentially improve bovine in vitro oocyte maturation is by extending the period before meiosis resumption. This will, conceivably, improve cytoplasmic maturation. This strategy is referred to as simulated physiological oocyte maturation (SPOM). It involves supplementing cultured COCs with cell-permeable cAMP agonists (e.g., forskolin, isobutyl-1-methylxanthine, cilostamide) to prevent meiosis resumption. This procedure was developed to improve oocyte quality by permitting longer periods of oocyte exposure to cumulus cells and hormones in the IVM medium before meiosis resumes [58]. There is a broad range in the duration of cAMP agonist supplementation (2 to 24 h treatment period in most studies) [59]. A recent meta-analysis determined that only 25% of SPOM studies utilizing bovine COCs identified a beneficial effect on subsequent blastocyst rates after removal from treatment and completing oocyte maturation and fertilization and embryo culture [59]. These observations suggest that SPOM is only effective in some circumstances. Unfortunately, the variables that dictate when SPOM will be effective at improving oocyte quality have not been defined. 

## 3. Fertilization

The maturation events described above yield an oocyte that is arrested at metaphase II (MII). This second meiotic arrest is maintained by elevated levels of cyclin-dependent protein kinase 1 (CDK1) activity in the oocyte [24]. Fertilization is the process by which oocytes and spermatozoa fuse so that their haploid chromosomes may combine and generate a new diploid organism. Ovulation occurs 24–30 h after the surge in LH in cattle, and the infundibulum of the oviduct will capture the ovulated COC and direct it into the oviduct. Fertilization occurs soon thereafter at the ampullary–isthmic junction of the oviduct. Spermatozoa need to undergo a process termed as capacitation before they can fertilize the oocyte [60]. This event alters sperm membrane architecture and permeability, modifies flagella activity, and removes surface proteins which allow sperm to bind to the zona pellucida (ZP) [60]. Glycoproteins, apply named ZP proteins, are responsible for sperm binding. This binding triggers the spermatozoon to undergo the acrosome reaction. This reaction is essential for the spermatozoon to gain the hyperactive motility needed to penetrate through the ZP by a combination of physical force and protease-dependent ZP digestion [61]. Sperm–oocyte fusion then ensues upon the formation of sperm–egg receptor complexes that include IZUMO-JUNO and Fertilin β (ADAM2)-integrin/disintegrins [62,63]. IZUMO-JUNO binding is interesting to mention because this fusion produces cytosolic Ca^2+^ oscillations, which induces meiosis resumption [64]. These Ca^2+^ pulses occurring with sperm–oocyte fusion also induce the release of cortical granule into the perivitelline space [65,66]. The enzymes released include those that modify ZP proteins in ways that prevent further sperm binding and penetration. Female and male pronuclei are formed soon after sperm–oocyte fusion and the completion of meiosis. These pronuclei increase in size, and then the nuclear envelopes are degraded and the sperm and oocyte genomes are conjoined [67]. This event is termed syngamy. 

Polyspermy is one parameter that can be problematic in bovine IVP systems. There is a direct association between sperm concentrations and polyspermy rates during IVF [68]. Polyspermy is also influenced by heparin concentrations in bovine IVF media [69]. Bull-by-bull optimization of both sperm numbers and heparin concentrations is usually required to maximize the fertilization potential while limiting polyspermy [70]. IVP embryo systems in the human, horse, and several endangered species use an intracytoplasmic sperm injection (ICSI) to prevent polyspermy as well as to optimize fertilization capacity when sperm quality may be limited or sperm numbers are in short supply [71,72]. Parthenogenesis is another concern in IVP embryo systems. The development of a female embryo without fertilization can be induced with various chemicals, and specifically those that interfere with cytoplasmic calcium availability and the activity of mitogen-activated protein kinases and serine–threonine protein kinases [73,74]. However, excessive ethanol in IVM medium (>3.0% *v*/*v*) and longer-than-normal IVM or combined IMV-IVF periods (e.g., >26 h or >44 h, respectively) will increase parthenogenesis rates [75,76,77]. 

Emerging tools are being developed to assist with quantifying oocyte and zygote quality. One already mentioned is BCB staining as an indicator of oocyte maturation. A few other of interest include correlations of oocyte and zygote quality with certain ZP characteristics, such as light refraction properties and thickness [74,78,79,80], and migration through a di-electrophoretic field [81]. 

## 4. Initial Cleavages and Transition to Embryo Genome Activation

Following fertilization, the newly formed zygote will undergo its first cleavage division. This occurs approximately 23 to 36 h post-fertilization in the cow [82,83]. The two-cell embryos will continue to undergo cell divisions, first resulting in a 4-cell (~42 h), then 8-cell (~60 h), and then 16-cell embryo (~102 h) [83]. A small amount of embryonic transcription can be detected in cow embryos as early as the two-cell, stage but the primary means for maintaining embryo viability and development at this time is through the translation of maternal mRNAs [31,84,85]. The transition from maternal to embryo genome activation (EGA) is associated with a developmental arrest (also known as developmental block) that was first characterized as a “2-cell block” in mice and later as a developmental arrest at the 8- to 16-cell stage in the cow [86,87,88,89,90]. Overcoming this developmental block provided a noteworthy achievement that allowed in vitro embryo production to serve as a developmental model, and later as an assisted reproductive technology for several species, including the cow. 

The timing and sequence of blastomere cleavages is being used to predict the competence of IVP bovine, human, porcine, and murine embryos [91,92,93,94,95]. This technology is beginning to gain traction in cattle embryos with homemade and commercially available embryo incubation chambers with a real-time imaging capability (e.g., Miri^®^ Time-Lapse Incubator; Esco Medical Group, Kringelled, Denmark). Several studies have associated the timing and symmetry of bovine blastomere cleavages with the potential to form blastocysts [96,97,98,99] and predict pregnancy success [100]. A recent study observed gene expression changes in bovine embryos that underwent synchronous cleavages (two-, four-, eight-cell stages) versus asynchronous cleavages (three-, five-, seven-cell stages) during early development [101]. The asynchronous cleaved embryos were better able to achieve the blastocyst stage than synchronous cleaved embryos [101]. Further work is needed to explore whether the timing and/or synchrony of bovine embryo cleavages can predict pregnancy success. 

One shortcoming of IVP embryo systems in all farm animals and in humans is the high prevalence of embryos containing too few or too many copies of one or more chromosomes. Problems with chromosomal pairing, replication, and crossing over can occur during meiosis in either gamete, causing aneuploidy. Chromosomal anomalies may also occur during mitosis, and this can produce aneuploidy when it occurs at the one-cell stage, or a mosaicism referred to as mixoploidy if it occurs on or after the two-cell stage. Aneuploidy and mixoploidy incidences range from 5% to 25% in IVP bovine embryos [102,103,104]. Only ~5% of these embryos will reach the blastocysts stage [102,103], and pregnancy rates from these embryos are very low (<5%) [102,103]. This outcome is not surprising given the lethal nature of these chromosomal abnormalities. However, it is interesting that normal, diploid embryos yielded a 60% pregnancy rate [103]. Thus, limiting the incidence of aneuploidy and mixoploidy and/or screening for these anomalies before ET can be an effective way to improve pregnancy success when using IVP embryos. Embryo screening may not be too far from being realized in cattle if the current trend of genomic testing in IVP bovine embryos continues [102,103,104]. 

## 5. Compaction and Blastocoel Cavity Stages of Development

The 16-cell bovine embryo is termed an early morula when individual blastomeres become difficult to distinguish from one another [105]. The next round of blastomere divisions produces a 32-cell stage compact morula given its tight compaction. This compaction creates an apical–basal polarity with an outer and inner group of blastomeres that will eventually differentiate into trophectoderm (TE) and the inner cell mass (ICM). This leads to the activation of RhoA, which causes polarization of the actin network and recruitment of the Par3-Par6-aPKC complex to the apical domain [106,107]. In addition, E-cadherin, a Ca^2+^-dependent adhesion molecule, drives compaction and proper apical–basal domain separation [108], and this leads to the formation of adherens junctions, gap junctions, tight junctions, and desmosomes within each region [109]. 

Cell junctions are especially important for the outer cells, where tight junctions are reinforced by actin ring structures form a lowly permeable seal around the embryo [110,111]. These actin rings first appear in the apical domain of the outer cells of the embryo, and then they expand to cell–cell junctions where they ‘zipper’ together, creating a stronger seal [111]. This permeability seal is necessary for an eventual influx of sodium ions, which is enabled by Na+/K+-ATPase pumps located on the basolateral domain of outside cells [112]. The osmotic gradient created in combination with aquaporins aids in water movement into the newly formed cavity [113]. The osmotic fluid first travels through several microlumens, which were formed by exocytosis of vacuoles and vesicles from the basal domain of outside cells [107,114]. Under increasing tension and pressure, the microlumens eventually combine into a singular cavity, termed the blastocoel cavity [115,116]. The filling of this cavity assists with microscopically distinguishing the outer from the inner cells, which are undergoing cell differentiation in the TE and ICM lineages, respectively, as blastulation occurs [107].

Several laboratories are exploring ways to improve IVP efficiency by supplementing culture medium with bioactive factors normally found in the reproductive tract during early pregnancy. These compounds are referred to as “embryokines” because of their potential to improve embryo production efficiency and/or embryo quality [117]. A recent review by our group provides a detailed overview of these embryokines [8]. Several embryokines have been described to improve blastocyst development by 5–10%, promote embryo cryo-survivability by 5–10%, and modulate TE and ICM development by 10–50% in IVP bovine embryos [8]. Thus far, three embryokines have improved pregnancy and/or calving success by as much as 15% in transferred IVP embryos [8]. These are colony stimulated factor 2 (CSF2) [118,119], Dickkopf WNT signaling pathway inhibitor 1 (DKK1) [119,120], and insulin-like growth factor 1(IGF1) [121]. 

## 6. First Embryonic Cell Lineage Segregation: ICM or TE

Compaction during the morula stage provides a means for establishing cellular polarity. This function is needed for the determination of cell fate identity. The Hippo signaling pathway is a key regulator controlling the cell fate decision of ICM versus trophectoderm (TE) at the blastocyst stage [122,123]. The Hippo signaling pathway controlling ICM versus TE specification was first investigated in the mouse. As discussed previously, the apical domain of outer cells is polarized with an actin network. This apical actin network sequesters a key Hippo signaling factor, angiomotin (AMOT), so it is unable to activate its Hippo signaling target, large tumor suppressor (LATS) [124,125]. This prevents the phosphorylation of Yes-associated protein 1 (YAP1) and its PDZ-binding motif, TAZ [126,127]. Non-phosphorylated YAP1/TAZ complex will translocate to the nucleus and regulate the genes involved in specifying TE cell fate, including TEA domain-binding transcription factor 4 (*Tead4*), which in turn stimulates the expression of Caudal-related homeobox transcription factor 2 (*Cdx2*) and GATA-binding protein 3 (*Gata3*) [125,128]. By contrast, the inner cells are devoid of an apical domain, and the absence of an apical actin network permits AMOT to interact with LATS to phosphorylate YAP1/TAZ, thus preventing their entry into the nucleus [125]. The absence of *Tead4*, *Cdx2*, *Gata3*, and other TE-specifying genes prevents these inner cells from differentiating into TE. Instead, they maintain a multipotent state capable of developing into any of the three germ layers and other extraembryonic membranes [125,129]. 

These ICM and TE specification events are similar in the cow embryo, but some noted mechanistic distinctions exist. These differences have been summarized nicely in a recent review [130]. Several of the Hippo signaling events mentioned earlier are the same for the cow, but for the cow there are noteworthy differences in the localization of both YAP1 and TAZ in cow embryos [122,131]. Specifically, phosphorylated YAP1 (pYAP1) is detected in both cytoplasmic and nuclear compartments in pre-compaction bovine embryos, and pYAP1 is predominately localized to the nucleus in post-compacted embryos [132]. Additionally, TAZ localization is less distinctive in the cow embryo. It is localized in the cytoplasm and in some but not all nuclei during the pre-compaction period in cattle but is exclusively localized to the nuclei in pre-compaction-stage mouse embryos [132]. The consequences of these changes in the lineage specification have yet to be fully resolved, but it is now becoming clear that some of the regulators of TE formation and development in the cow embryo may be different from what has been observed in the mouse. For example, TEAD4 does not appear to affect the expression of key TE-specifying genes, including CDX2 and SOX2, similar to how it does in the mouse embryo [133]. 

Several laboratories, including our own, have been keenly interested in exploring how to increase the number of ICM and TE cells, with the overarching goal of improving IVP bovine embryo quality. Several embryokines appear to influence TE cell proliferation (IGF1, follistatin [FST], growth hormone, melatonin), and another set of embryokines will influence ICM cell numbers (FST, interleukin-6, DKK1, CSF2) (reviewed in [8]). Efforts are also being made to identify specific cocktails of embryokines that will improve IVP embryo quantity, quality, and cryo-survivability [57,134,135,136,137]. Further work combining the mechanistic features of embryokines with lineage specification events could one day permit us to produce IVP embryos with ICM and TE cells that more closely resemble these cells found within in-vivo-generated embryos. 

## 7. Second Cell Lineage Segregation: PE or EPI

The ICM differentiates into two cell lineages within a few days after the blastocyst has formed. These cell types are defined as primitive endoderm (PE) and the epiblast (EPI). The PE is also referred to as the hypoblast (HYPO) when they migrate to the base of the ICM and migrate around the base of the TE layer to form the yolk sac [138,139]. The EPI cells will develop into the three germ layers (i.e., ectoderm, mesoderm, and endoderm) and the extraembryonic mesoderm, which produces the allantoic sac [140,141]. These developmental events will be discussed later in this review. 

Several transcriptional regulators facilitate this cell fate determination [142], but we will focus our discussion on two factors that have been used as markers for EPI and PE lineages. These are GATA-binding protein 6 (GATA6), which promotes PE emergence, and NANOG homeobox (NANOG), which is maintained in EPI cells [140]. Both factors are expressed in all ICM cells in the early blastocyst, but within 1–2 days, depending on the species, the ICM begins to exhibit a random “salt and pepper” distribution pattern of ICM cells preferentially expressing GATA6 or NANOG [140]. Selective apoptosis and cell movement enables GATA6^+^ cells to migrate to the base of the ICM, and NANOG^+^ cells to remain within the original region of the ICM, positioned above the HYPO region [140,143]. Various PE and EPI markers have been used to establish this process, and it is now becoming more common to see GATA4 in place of GATA6 when examining PE development [144]. 

The process of PE-EPI specification was initially described in the mouse. Fibroblast growth factor (FGF)/MAPK signaling is the key regulator of this process [145,146]. The disruption of either *Fgf4* or its cognate receptor, *Fgfr2*, produces an early post-implantation embryo lethality characterized by the absence of a yolk sac in the mouse [147,148]. Embryo-derived FGF4 signaling through FGFR2 is now established as the controller of PE and EPI specification [146]. The decision for a PE versus EPI differentiation appears to occur arbitrarily, and those cells with a high abundance of FGFR2 are more likely to develop into PE cells, whereas those cells that contain less FGFR2 but express more FGF4 will develop into EPI cells [146,149,150,151]. There is plasticity early in this process, and supplementing FGF4 or a homolog, FGF2, will skew differentiation towards more PE cells, and inhibiting FGFR2 or MAPK signaling will prevent PE development and promote ICM cells into the EPI lineage [139,145]. Another FGF receptor that recognizes FGF2 and FGF4, termed FGFR1, is also involved with PE-EPI specification [150]. This receptor is expressed in all ICM cells, and its MAPK-stimulatory actions mimic FGFR2 in PE-developing cells, whereas its MAPK-dependent activities in EPI-developing cells aid in conferring EPI identity by stimulating *Nanog* expression [150]. 

Many of the same mechanistic control points for EPI-PE determination observed in the mouse also occur in the cow. Both FGF2 and FGF4 supplementation promotes PE development and limits EPI development in bovine embryos, and blocking FGFR2 or MAPK activity limits PE development [152,153]. However, there are some intriguing differences in this specification event observed in cow embryos. NANOG is not required for PE development in the mouse, but it does appear necessary to maintain high GATA6 expression within PE cells in cow embryos [154]. In addition, MAPK inhibition will not completely prevent PE development in cow embryos, similar to how it does in mouse embryos [152,155,156]. These distinctions in cell fate processes suggest that the cow embryo contains additional mediators of PE-EPI determination. These mediators remain unknown, but recent work in our laboratory identified an additional strategy that bovine embryos may employ to control the relative numbers of PE and EPI cells. We observed that IL6, an embryo and uterine-derived cytokine, and LIF, a uterine-derived cytokine, promote PE development after PE has been specified using a Janus-kinase-dependent mechanism [157,158]. More work is required to determine the importance of this and related control points in PE-EPI development in cattle, mice, and other mammals.

Understanding the molecular mechanisms that govern cell lineage segregation into EPI and PE is important because any abnormalities in these processes can negatively affect pregnancy retention. A subset of pregnancies generated from bovine IVP embryos exhibit various morphological abnormalities, and one of the most notable abnormalities is poorly developed yolk sacs [159,160]. The yolk sac is the primary controller of nutrient consumption and processing until day 30–40 of gestation [161,162]. The EPI is also influenced by in vitro culture. A recent report identified several IVP-dependent alterations in the EPI transcriptome and epigenetic patterning associated with embryonic development defects [163]. It is unclear why some IVP embryos fail to generate a fully functioning yolk sac or a properly programmed EPI. However, it is reasonable to suspect that improvements in ICM development will facilitate adequate yolk sac development and vascularity and ensure appropriate EPI development. 

## 8. Elongation

The cow and other ungulates (e.g., sheep, deer, and pigs) are very different from many other species in the timing and invasiveness of implantation. It is distinctly different from primates and rodents, where blastocysts invade into the uterine wall soon after hatching [164]. Ruminant blastocysts remain free-floating for multiple days after hatching. At day 13 to 14 of development in the cow, TE cells will flatten and undergo rapid hyperplasia, and this initiates an elongated and eventually filamentous shape [4]. This structure is referred to as a conceptus (complete assemblage of embryonic and extraembryonic tissues) [165]. Elongation allows increased surface area contact with the uterus in preparation for uterine attachment. It also facilitates an extensive network of conceptus–endometrial crosstalk that is essential for the normal progression of pregnancy (see [4] for recent review). The bovine conceptus will continue to elongate until day 19–21, when it has completely attached to the uterine wall [4]. 

Other reviews have indicated that failures in conceptus elongation are prominent in lactating dairy cattle and suckling beef cows [1,2,9]. Unfortunately, performing research during the elongation period is more challenging than earlier stages of bovine embryogenesis because no bona fide filamentous conceptus culture systems exist for ruminants. Short-term cultures of elongated and filamentous conceptuses recovered from cattle have proven useful for understanding transcriptomic regulators of the elongation process (reviewed in [4]). Several bovine TE lines are also available to study some of the TE actions occurring during the elongation period [135,166,167]. Bovine trophoblast stem cell lines are also available, and these cells have the capacity to proliferate as mononucleated TE or differentiate into binucleated TE [168]. Lastly, recent strides have been made in exploring the origins of EPI-PE differentiation using extended cultures with media formulations used in primed embryonic stem cell (ESC) and induced pluripotent stem cell (iPSC) culture systems [169,170,171]. These media formulations permit blastocysts to undergo complete HYPO migration while promoting EPI cell survival, albeit at a slower rate than what occurs in vivo. Thus, although elongation is not occurring, it certainly appears that we are learning how to extend in vitro EPI and HPYO survivability. 

## 9. Gastrulation

Gastrulation is a process that involves a series of cell differentiation and migration events that produce multiple germ cell layers and lays out what is referred to as the embryonic disk (i.e., the embryo proper) into the basic anterior–posterior and dorsal–ventral axes [172]. These events begin soon after EPI-PE specification and are loosely concurrent with the initiation of elongation [173,174]. One of the first events observed in the cow is the programmed cell death of TE cells in contact with the EPI-HYPO region, a structure termed the embryonic disk. These TE cells are termed the Rauber’s Layer, and their removal occurs in cattle, pigs, horses, and several other species but not in others, including humans and rodents [175]. A recent report [175] suggests that the removal of the Rauber’s layer in some species and not others is a consequence of various gastrulation signaling factors (i.e., NODAL, BMP, WNT). The absence of a Rauber’s layer limits the signal strength of these gastrulation signals, thus preventing an overgrowth of EPI and germ layers. Embryos that maintain their Rauber’s layer have developed an alternative strategy to distance themselves from these signals by creating an EPI cavity that forms the basis for the amnionic cavity.

A series of cell lineage emergences and cell movement occurs between day 11 and 14 of development. The embryonic disk contains EPI and HYPO layers at this early stage of development. This is referred to as the bilaminar disk. Gastrulation begins around day 11 in the cow, with EPI cell proliferation and Rauber’s layer disappearance. At day 12, the anterior region of the HYPO, termed the anterior visceral hypoblast (AVH), produces fine projections into the EPI cells [173]. These AVH projections cause EPI cell accumulation at the posterior end of the bilaminar disc at a location that will become the site of the primitive streak [176]. After the disappearance of Rauber’s layer, the EPI cells form the three germ layers. The first of these layers is the embryonic ectoderm [173,177,178]. The arrangement of ectoderm at the posterior end of the embryonic disk initiates primitive streak formation around day 14 in cattle [179,180]. As embryonic ectoderm is migrating towards the center of the embryonic disc, the embryonic endoderm will inhabit the base of the embryonic disk, replacing the HYPO [173]. Other EPI cells will form the embryonic mesoderm, and these cells will migrate laterally between the EPI and HYPO [180]. These cell movements and differentiation events form what is referred to as trilaminar. Neurulation and organogenesis will commence soon after gastrulation is complete. 

Problems with EPI development and the proper completion of gastrulation exist in IVP embryos. The size and morphology of the ICM are primary morphological indicators of bovine and human IVP embryo quality [181,182]. Additionally, it is well established that bovine IVP blastocysts have fewer total ICM cells than their in-vivo-produced counterparts [183,184,185,186]. This likely leads to a reduction in the size of the embryonic disk. Indeed, embryonic disk size, but not overall conceptus size, is reduced in day 17 IVP conceptuses when compared with in-vivo-produced conceptuses [187]. Several research groups have noted that embryonic disks cannot be detected via stereomicroscopy in a subset of day 15–17 IVP bovine conceptuses [188,189,190,191,192]. The IVP conceptuses with undetectable embryonic disks were less likely to retain pregnancies than IVP conceptuses with visible embryonic disks (62.5 vs. 6.7%, detectable vs. nondetectable embryonic disks, respectively) [189]. 

Improvements in embryo culture seem to be minimizing the adverse effects for IVP-derived embryonic disks. Attempts to encourage bovine conceptus elongation by culturing embryos in agarose tunnels failed to maintain embryonic disk survival [177,193,194]. However, more recent investigations into modifying IVP embryo media composition offer encouraging insights into how to improve embryonic disk survival. Supplementation with insulin, transferrin, and selenium improved embryonic disc development in transferred day 13 transferred IVP embryos [195]. Similarly, supplementing prostaglandin E2 during in vitro oocyte maturation improved embryonic disc development in transferred day 15 bovine conceptuses [196]. Stem-cell-media-based systems are also being used to generate certain components of gastrulation in cultured bovine and ovine embryos [169,197,198]. Specifically, markers for mesoderm and primitive streak formation were observed in day 14 IVP sheep embryos [198]. A temporal delay exists in this culture system, and day 14 IVP sheep embryos resemble day 11 in-vivo-derived sheep embryos. 

## 10. Early Placentation

The development, attachment, and limited migration of the bovine placenta has been reviewed recently [4], so the topic will be discussed briefly, then we will proceed with describing how we are using information about early placental development to better understand why some pregnancies fail. As discussed earlier, the apposition of TE cells to the endometrial epithelium begins at day 19–20 of gestation. A fusogenic TE cell type, referred to as the binucleate cell (BNC), given that it contains two distinct nuclei, will fuse with uterine luminal epithelium to generate a hybrid, tri-nucleated cell [199,200]. Further fusing and cell lysis creates a syncytium layer that is evident for a few weeks in cattle before returning to a state where TE invasion is primarily limited to hybrid trinuclear cells [201]. This short period of extended syncytium formation is associated with the period of gestation when the placenta is defined as a choriovitelline placenta, where it utilizes the yolk sac as the digestive structure to the capture and process the nutrients derived from uterine histotroph and cell debris [3,202,203]. Around day 30 to 40 of gestation, the placenta begins to function primarily as a chorioallantoic placenta, where it utilizes the nutrients derived from placentomes created by the penetration of fetal cotyledonary villi into uterine caruncular crypts [3,4,204]. 

Differences exist in placentae from in-vivo- or in-vitro-produced pregnancies. In sheep, placentae from IVP pregnancies were smaller in size and weighed less than their control placentae counterparts [205]. Likewise, cow placentomes were thinner yet longer in IVP-generated pregnancies [189,206]. This increase in placentome length is intriguing because it may represent a compensation mechanism to maintain normal nutrient and waste flow within the uterus. These and other IVP-dependent modifications in placental structure and function have been implicated in producing large gestational age calves, a phenomenon commonly referred to as large offspring syndrome [11,207,208]. However, the prevalence and severity of this syndrome has become less evident in recent years, likely due to modifications in media formulations and culture conditions (e.g., removal of serum, low oxygen environment) [3,9,209]. 

## 11. Concluding Remarks

This review explored when and why pregnancy losses associated with embryonic factors occur in the first month of gestation in cattle. We have outlined and explained a variety of problems that can arise during oocyte maturation, fertilization, early embryonic development, compaction and blastulation, embryonic cell lineage specification, elongation, gastrulation, and placentation. It is not surprising that pregnancies will usually terminate when problems in these early embryological events occur. The high prevalence of these early gestational pregnancy losses is costly to the dairy and beef industries. Efforts to limit these losses require a continued exploration of the key events of early embryogenesis. The IVP bovine embryo is an excellent model for much of this research. Unfortunately, current IVP systems also contain limitations, one of which is that the embryos derived from these systems are even less competent to survive after transfer than in-vivo-derived embryos. This has created an area of research with the objective of creating IVP embryos with a greater likelihood of surviving post-transfer. Several remediation strategies have been provided herein to improve overall IVP embryo competence, but it is readily evident that more research is needed to improve IVP embryo quality. 

## Figures and Tables

**Table 1 animals-13-01760-t001:** Limitations with in vitro embryo production and potential remediation schemes for oocyte maturation, fertilization, and early embryo development.

Facet of Development	Limitation	Remediation
Oocyte Maturation	Non-lethal assessment of competency Inefficient nuclear maturation	BCB staining Supplement growth factors and cytokines found in the follicle
	Incomplete cytoplasmic maturation	Simulated physiological oocyte maturation (SPOM)
Fertilization	Polyspermy	Optimize IVF on a bull-by-bull basis ICSI
	Parthenogenesis	Heparin concentration in IVF medium >26 h IVF or >44 h IVM+IVF
Initial cleavages	Aberrations in the timing and sequence of initial cleavages	Real-time imaging of cleavage stage development
	Aneuploidy/mixoploidy	Couple ploidy assessments with embryo genomic testing
Compaction and cavitation	Suboptimal development to morula and blastocyst stages	Embryokine supplementation

**Table 2 animals-13-01760-t002:** Limitations with in vitro embryo production and potential remediation schemes to bovine embryos during and after blastocyst formation.

Facet of Development	Limitation	Remediation
ICM:TE specification	Poor blastocyst quality Low ICM and/or TE cell numbers	Learn more about ICM:TE specification Embryokine supplementation
EPI:PE specification	Incomplete understanding of this specification event in cattle	Learn more about PE:EPI specification
Conceptus elongation	Lack of *bona fide* filamentous conceptus culture systems	Refine culture conditions
Gastrulation	Poor/delayed progression Marginally effective in vitro culture system models	Improve embryo culture conditions Further optimize these systems
Early placentation	Poor yolk sac development Altered placentome development	More research into bovine yolk sac development More research into TE development/differentiation

## Data Availability

No new data were created for this review.

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
