# Peer review of "Developmental Hurdles That Can Compromise Pregnancy during the First Month of Gestation in Cattle"

_animals, 2023, doi:10.3390/ani13111760_

Round 1

Reviewer 1 Report

This review is a very comprehensive overview.  It does not attempt to go into the mechanistic details but does provide appropriate citations if the reader wishes to go more in depth. Thus, it will provide a useful guide to students and researchers who are entering the field or wish to broaden or refresh their understanding.

I did note a few typos as follows.

line 122:  ability [t tacked on to the end of the work]

line 273: osmotic not osmatic

line 384: Il6...and LIF... promote, not promotes

line 398: (e.g., sheep, deer, and pigs).   A list needs to include "and".

line 426: extra space between "lines" and "are"

line 437: (i.e., embryo-proper).  Add comma after i.e. and maybe add "the" (i.e., the embryo-proper)

line 536: within - the cumulus-oocyte-complex within the follicle

Author Response

Thank you for the kind words. We are excited that you think this review has merit. 

And thank you for catches these various spelling mistakes and added spaces.  We have made all the modifications you indicated. 

Reviewer 2 Report

Line 27: Missing events between Day 0 and Day 7 ike genome activation shouldn't be addedd to the list?

Not only "events" associated to the embryo, induce pregnancy losses during the first weeks of gestation. How about uterine environment during the first week of pregnancy when the embryo arrives into the uterus.? Environmental factors? ect ect.  I would suggest to add to the list all the potential factors that induce pregnancy lossess during the first month of gestation, and then clarify that this paper will focus on embryonicfactors.

Line 43:  Aren't OPU techniques part of IVP embryo systems? Wouldn't be better to say "..are combining embryo systems using OPU techniques?

Line 45: Is that the last IETS version? Please update and rephrase the sentence

Line 52: That's a big statement but unfortunately the numbers are pretty much the same as 20-30 yrs ago. If you check references from 90's, 30% has been the expected % of blastocyst development in IVP embryos and quite substatntial research work have been done in this area. Not sure if that statement is valid.

Line 68: I don't think that the discussion is focus on the cow. In my opinion, the focus is entirely in embryonic issues, but maternal factors (i.e uterine environement) are not consider at all in the text. 

Line152: I will add somewhere how much of the IVM oocytes did not get nuclear/citoplasmic maturation. Also, oocyte maturation is normally assessed by cumulus cell expansion. Would you suggest another approach to measure maturation success rather than morphological -subjective-assessments?

Line 158:  How long? range?

Line 160: Is 2-24 hr the " extended" period mentioned above?

Line 244: what %?

Line 251: This part of mathematical models should be moved somewhere else but not at the end of the paragraph. I think this whole paragraph needs a conclusion.

Table 1: Fertilization row, Remediation Column: Could you be more specific in the composition and IVM duration's to avoid parthenogenesis

Line 290:  Even though the previous review in your group mentioned the embryokines, could you please summarize for this reviewt, the blastocyst production &pregnancy rates obtained after using these embryokines in the culture medium.

Line 335: Not sure if the last sentence from " Further work combining...." should be in that section or maybe moved to the Conclusions.

Line 426: delete the space

Line 433: Following the structure of the previous sections, I am missing in this section the description of the potential causes of poor/lack of elongation. Even though the text has been focused mostly in IVP embryos, I think elongation issues should be addressed in larger extent as being one of the main causes of failure of  in cattle.

  Line 438: delete the space

Line 505: Could you mention specifically what strategies are mentioned to avoid poor placentation in IVP embryos?

Table 2: Remediation column: I think this column needs to be more specific. Saying "more research...Continue refining culture conditions.. Improve embryo conditions etc.." is not specific and its quite obvious. What are the specific startegies to overcome theses issues?

Line 510: This part (510-517) shouldn't be moved to LINES 392 onwards?

Line 519:  I will add.."why pregnancies losses associated to EMBRYONIC FACTORS", because you did not mentioned any other factors that might cause pregancy lossess during the first month of pregnancy. 

Line 526: "e" is missing  

Line 564: Avoid capital letters

Lines 601, 652, 669, 693, 708, 713, 730, 736, 790, 797, etc and throughout the references: Use either abbreviated of full name of the journals

Author Response

Thank you. These comments were addressed. 

ln 27: We added more to the introduction to mention how the oviduct/uterine contributes to pregnancy losses. Thanks for suggesting this. We think this is an important addition. 

Ln 43: changed. 

Ln 45: Changed. Thanks for catching this oversight. 

Ln 52: agreed. We rephrased things to tone down the statement. that said, one must remember that nearly all of the work completed 20+ years ago used serum in media, and the presence of serum will improve blast rates by 10-20%.  So, we are comparing apples and oranges if we compare today's efficiency with the efficiency several decades ago. 

Ln 68: Good pointy. We modified this text accordingly. 

Ln 152:  Again, nice point. We added some text to mention cumulus expansion. 

Ln 158 and 160: We made some adjustments to clarify the range in the treatment periods. 

Ln 244: Added this percentage (<5%). 

Ln 251: Agreed. We removed this sentence.  

Table 1. Changed. 

Ln 290: We added these percentages. 

Ln 335: We see your point, but this seems like the ideal place to make this conclusion, so we have not moved it.  

Ln 426: change made. 

Ln 433: Thanks for catching this oversight. We added a point about the importance of conceptus elongation and also rearranged some of the content. 

Ln 438: done. 

Ln 505: Examples have been added. 

Table 2: We still are unclear how to remediate these limitations, so we kept thing vague. 

Ln 510: Thanks for catching this duplication. We removed this paragraph and placed some of this information into the previous section you indicated. 

Ln 519: made the change. 

Ln 526: fixed. 

References: Made the modifications you specified.